# Balance dysfunction the most significant cause of in-hospital falls in patients taking hypnotic drugs: A retrospective study

Ryuki Hashida[1,2]*, Hiroo Matsuse[1,2], Shinji Yokoyama[3], Sayuri Kawano[4], Eriko Higashi[2], Hiroshi Tajma[1,2], Masafumi Bekki[1,2], Sohei Iwanaga[1,2], Koji Hara[1,2], Yosuke Nakamura[1,2], Yuji Kaneyuki[1,2], Takeshi Nago[1,2], Yoshihiro Fukumoto[3], Motohiro Ozone[5], Naohisa Uchimura[5], Naoto Shiba[1,2]

**1** Department of Orthopaedics, Kurume University, Kurume University School of Medicine, Kurume, Japan, **2** Division of Rehabilitation, Kurume University Hospital, Kurume, Fukuoka, Japan, **3** Division of Cardiovascular Medicine, Department of Internal Medicine, Kurume University School of Medicine, Kurume, Japan, **4** Department of Nursing, Kurume University Hospital, Kurume, Fukuoka, Japan, **5** Department of Neuropsychiatry, Kurume University School of Medicine, Kurume, Japan

* hashida_ryuuki@med.kurume-u.ac.jp

**Data Availability Statement:** All relevant data are available on the Harvard Dataverse Network: https://doi.org/10.7910/DVN/WTMVGP.

## Abstract

### Purpose

Preventing falls in patients is one of the most important concerns in acute hospitals. Balance disorder and hypnotic drugs lead to falls. The Standing Test for Imbalance and Disequilibrium (SIDE) is developed for the evaluation of static standing balance ability. There have been no reports of a comprehensive assessment of falls risk including hypnotic drugs and SIDE. The purpose of this study was to investigate the fall rate of each patient who took the hypnotic drug and the factor associated with falls.

### Methods

Fall rates for each hypnotic drug were calculated as follows (number of patients who fell/number of patients prescribed hypnotic drug x 100). We investigated the hypnotic drugs as follows; benzodiazepine drugs, Z-drugs, melatonin receptor agonists, and orexin receptor antagonists. Hypnotic drug fall rate was analyzed using Pearson's chi-square test. Decision tree analysis is the method we used to discover the most influential factors associated with falls.

### Results

This study included 2840 patients taking hypnotic drugs. Accidents involving falls were reported for 211 of inpatients taking hypnotic drugs. Z-drug recipients had the lowest fall rate among the hypnotic drugs. We analyzed to identify independent factors for falls, a decision tree algorithm was created using two divergence variables. The SIDE levels indicating balance disorder were the initial divergence variable. The rate of falls in patients at SIDE level $\leqq$ 2a was 14.7%. On the other hand, the rate of falls in patients at SIDE level $\geqq$ 2b was 2.9%. Gender was the variable for the second classification. In this analysis, drugs weren't identified as divergence variables for falls.

**Funding:** Disclosures: Ryuki Hashida, Sayuri Kawano, Masafumi Bekki, Sohei Iwanaga, Koji Hara, Eriko Higashi, Hiroshi Tazima, Yosuke Nakamura, Yuji Kaneyuki, Shinji Yokoyama, Takeshi Nago, Naoto Shiba declare that they have no conflict of interest. Hiroo Matsuse received lecture fees from SK-Electronics CO., LTD. Yoshihiro Fukumoto received lecture fees from Daiichi-Sankyo CO., LTD. Motohiro Ozone received lecture fees from Eisai Co., Ltd., Takeda Pharmaceutical CO., LTD and MSD K.K. Naohisa Uchimura received lecture fees from Eisai Co., Ltd., Takeda Pharmaceutical CO., LTD. and MSD K.K. The funders had no role in study design, data collection and analysis, decision to publish, or preparation of the manuscript.

**Competing interests:** We have declared that no competing interests exist.

**Abbreviations:** SIDE, Standing Test for Imbalance and Disequilibrium.

## Conclusion

The SIDE balance assessment was the initial divergence variable by decision tree analysis. In order to prevent falls, it seems important not only to select appropriate hypnotic drugs but also to assess patients for balance and implement preventive measures.

## Introduction

Falls and related injuries are important problems in acute hospitals. About 2% of hospitalized patients experience falls during their hospital stay [1]. In one study, among the hospitalized patients who experienced falls, 30% experienced mild injuries and 5% experienced severe injuries [2]. Hip fracture is considered one of the most severe injuries caused by in-hospital falls, due to complications reducing quality of life, increasing required staff hours, and incurring unnecessary social cost [3]. The patients with a hospital-acquired hip fracture are already in poor condition physically and mentally and have an especially poor outcome [4]. Thus, patients with a hospital-acquired hip fracture have worse outcomes including death and discharging to nursing care facilities than patients who fall outside of the hospital [5]. Moreover, falls in the hospital are detrimental to the patient and can result in medical lawsuits. Therefore, preventing falls in patients is one of the important concerns in acute hospitals.

Falls have multiple causes. Compromised physical function, including balance disorder and gait disorder, lead to falls as well as hypnotic drugs [6]. Standing balance is important in predicting a patient's risk of falling. The Standing Test for Imbalance and Disequilibrium (SIDE) has been developed for the evaluation of static standing balance ability [7]. SIDE classifies the patients into six grades, the classification of patients based on their abilities would facilitate communication between medical staff and the development of fall-prevention programs. We have started using SIDE in assessing hospitalized patients to prevent falls.

Some kinds of medications including hypnotics, antidepressants, anti-hypertensive drugs, and diuretics cause falls [8]. Among them, hypnotics are often ordered as a single-use solution for insomnia because insomnia is a common symptom in hospitalized patients. Benzodiazepines have been widely prescribed for treating insomnia, however, they have severe side effects such as rebound insomnia, delirium, dementia, and falls [9, 10]. To avoid these severe side effects, benzodiazepines aren't recommended to be prescribed for elderly patients [3], and melatonin receptor agonists and orexin receptor antagonists have been developed [11, 12]. However, in one study, even zolpidem and suvorexant resulted in impaired balance ability versus a placebo [12]. Therefore, it is crucial to take into account the patient's balance ability and prevent falls when prescribing hypnotic drugs. To the best of our knowledge, there is no study to investigate the relationship between balance ability and falls in hospital patients who took hypnotic drugs.

The purpose of this study was to investigate the fall rate of each patient who took the hypnotic drug and the factor associated with falls.

## Methods

### Ethics

This study protocol conformed to the ethics guidelines of the Declaration of Helsinki, as reflected in prior approval given by the institutional review board of Kurume University Hospital (approval ID:20167). Informed consent from patients was obtained using an opt-out approach.

## Patients

We did a retrospective study and enrolled all hospitalized patients who were taking hypnotic drugs from July to December 2020 at Kurume University Hospital. Our hospital has 1,018 beds in 23 medical departments. We summarized inclusion and exclusion criteria in Fig 1. We obtained data of falls from incident and accident reports submitted by medical personnel.

## Drugs

We investigated the top 6 hypnotic drugs that were often prescribed as follows; benzodiazepine drugs (brotizolam), Z-drugs (eszopiclone, zolpidem), melatonin receptor agonists (ramelteon), and orexin receptor antagonists (lemborexant, suvorexant). We extracted patient data from prescription records of inpatient hypnotic drugs. Therefore, the number of patients is the total number of patients. We also investigated the dosage of each patient.

   We investigated other drugs as follows, angiotensin converting enzyme inhibitors, alpha blockers, beta blockers, alpha beta blockers, loop diuretics, and selective serotonin reuptake inhibitors which were previously reported to be the cause of falls [8, 13]. We defined polypharmacy as patients who were taking three or more drugs [14].

## Evaluation of balance

The SIDE is useful to assess balance ability at the bedside by various medical personnel (Fig 2). The accuracy of SIDE is validated by comparing the results with another functional balance scale; the Berg Balance Scale [7]. We defined patients who were unable to do tandem standing as having balance disorder (SIDE level 0, 1, and 2a).

## Statistical analysis

Fall rates for each hypnotic drug were calculated as follows (number of patients who fell/number of patients prescribed a hypnotic drug x 100). Each hypnotic drug fall rate, age, and balance disorder were analyzed using Pearson's chi-square test. Logistic regression analysis was conducted to investigate falls in each hypnotic drug. Decision tree analysis is a data mining

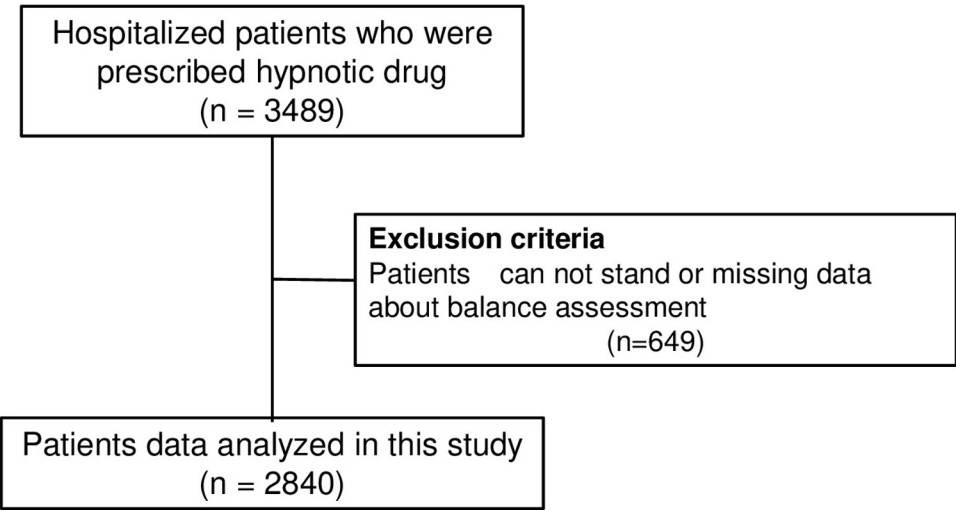

**Fig 1. Diagram of patient's inclusion and exclusion criteria in this study.** All patients admitted to our hospital from July to December 2020 who were prescribed hypnotic drugs were included in the study. Finally, 2840 patients were included in the analysis.

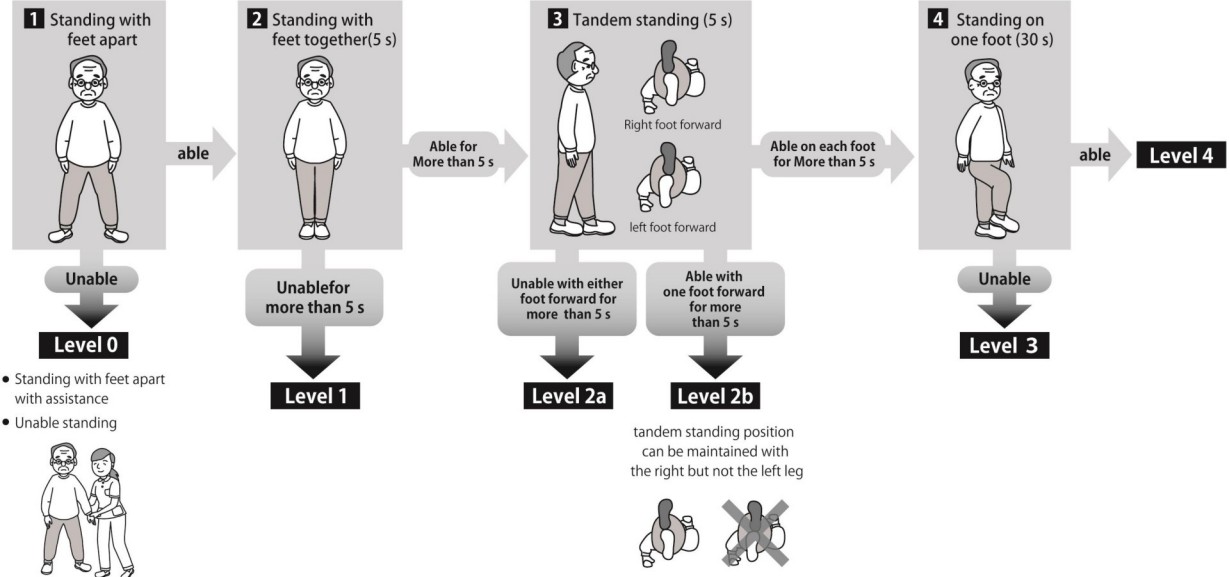

**Fig 2. Standing Test for Imbalance and Disequilibrium (SIDE).** The patients perform the static balance test in the following order, standing with feet apart, standing with feet together, tandem standing, and standing on one foot. After checking which movements are possible or impossible for the patient, the medical staff assesses the patients into six levels: Level 0, 1, 2a, 2b, 3, and 4. Level 0: The patient can't stand with feet apart without assistance. Level 1: The patient can stand with feet apart without assistance. However, the patients can't stand with feet together for more than 5 s. Level 2a: The patient can stand with feet together for more than 5 s. However, the patients can't stand in a tandem position with either foot forward. Level 2b: The patient can stand tandem with one but not the other foot in the leading position for more than 5 s. Level 3: The patient can stand tandem standing with each foot forward for more than 5 s. However, the patient can't stand on one foot more than 30 s. Level 4: The patient can stand on one foot more than 30 s with either foot.

technique that identifies high-priority factors [15]. Decision tree analysis reveals factors even if no a priori hypothesis is imposed. Therefore, decision tree analysis helps to discover the most influential factors associated with falls. This approach is useful to discover hidden factors that cannot be identified by logistic regression analysis. Decision tree analysis can identify the combinations of factors that constitute the highest risk for a condition of interest [16]. Therefore, a decision-tree algorithm was constructed to reveal profiles associated with falls. Subjects were classified according to the indicated cut-off values of the variable. A p-value <0.05 was considered statistically significant.

## Results

### Characteristics of patients and falls rates

This study included 3489 patients, 649 patients were excluded because they couldn't be assessed by SIDE or data was missing. Finally, 2840 patients were involved in this study. Median age of patients was 70.5 (59–77.9) years-old. Gender ratio of patients was male/female (55% / 45%). Falls incidents occurred in 27 departments, and the most prevalent department was Respiratory M edicine (13.1%) (Table 1). The drugs prescribed to the patients were as follows: benzodiazepine drugs; 276 (9.7%), Z-drugs; 722 (50.3%), melatonin receptor agonists; 413 (14.6%) and orexin receptor antagonists; 722 (25.4%). The SIDE levels of patients were level 0 (n = 352, 12.4%), level 1 (n = 186, 6.5%), level 2a (n = 542, 19.1%), level 2b (n = 543, 19.1%), level 3 (n = 510, 18.0%), and level 4 (n = 707, 24.9%). The number of patients who were assessed by SIDE to have a balance disorder was 1080 (38.0%).

**Table 1. Prevalence by department.**

|  | % | n |
|---|---|---|
| Respiratory Medicine | 13.10 | 372 |
| Neuropsychiatry | 9.79 | 278 |
| Orthopedic Surgery | 9.44 | 268 |
| Digestive Surgery | 8.20 | 233 |
| Cardiovascular Surgery | 7.78 | 221 |
| Cardiovascular Medicine | 6.23 | 177 |
| Otolaryngology | 6.16 | 175 |
| Urology | 4.82 | 137 |
| Neurosurgery | 4.30 | 122 |
| Gastroenterology | 3.59 | 102 |
| Gynecology | 2.99 | 85 |
| Nephrology | 2.99 | 85 |
| Hematology | 2.82 | 80 |
| Ophthalmology | 2.71 | 77 |
| Dermatology | 2.32 | 66 |
| Palliative care | 2.04 | 58 |
| Internal medicine of endocrinology and metabolism | 1.69 | 48 |
| Respiratory Surgery | 1.62 | 46 |
| Medical Oncology | 1.48 | 42 |
| Radiology | 1.41 | 40 |
| Emergency Medicine | 1.34 | 38 |
| Plastic Surgery | 1.30 | 37 |
| Oral Surgery | 1.20 | 34 |
| Breast Surgery | 0.49 | 14 |
| Pediatric Surgery | 0.11 | 3 |
| Pediatrics | 0.07 | 2 |

## Fall rates for each hypnotic drug

Fall accidents were reported for 211 (7.4%) of 2860 inpatients taking hypnotic drugs. The fall rate for each hypnotic drug was as follows: Z-drug (5.5%); melatonin receptor agonist (7.5%); benzodiazepine drugs (8.7%) and orexin receptor antagonists (10.7%) (Fig 3). The fall rate for z-drugs was significantly lower than for benzodiazepine drugs, and orexin receptor antagonists (p = 0.0432, p < .0001). However, there was no significant difference in the fall rate between z-drugs and melatonin receptor agonists (p = 0.1352).

## Age, and balance disorder in each hypnotic drug

Patients who were prescribed melatonin receptor agonists and orexin receptor antagonists were significantly older than those receiving z-drugs (p < .0001, p < .0001). There was no significant difference in age between patients who were prescribed z-drugs and benzodiazepines drugs (p = 0.1831) (Fig 4A).

Ratio of balance disorder in each hypnotic drug were shown in Fig 4B. Ratio of balance disorder in melatonin receptor agonists and orexin receptor antagonists were significantly higher than those of z-drugs (p < .0001, p < .0001). There was no significant difference in balance disorder ratio between z-drugs and benzodiazepine drugs (p = 0.3819) (Fig 4B).

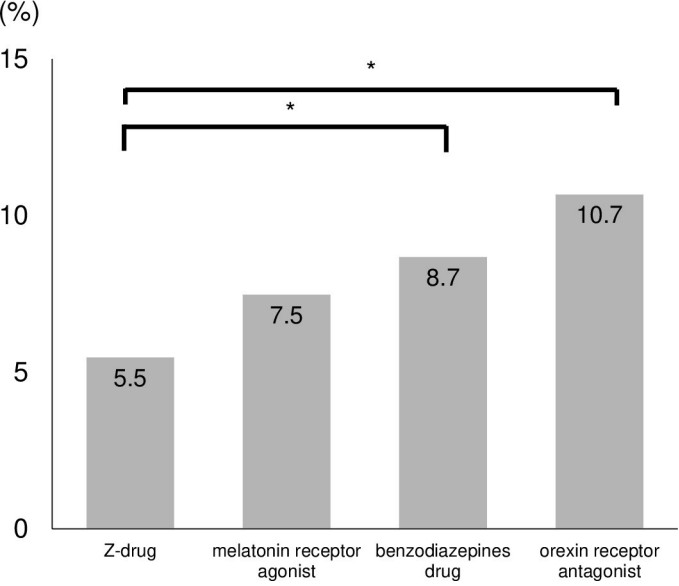

**Fig 3. Fall rate for each hypnotic drug.** Fall rate for z-drugs was significantly lower than those for benzodiazepine drugs and orexin receptor antagonists. *: means statistically significant.

## Cox regression analysis for falls

Z-drugs were significantly associated with a decreased odds ratio of falls in the univariate analysis (p < .0001). Orexin receptor antagonists were significantly associated with an increased

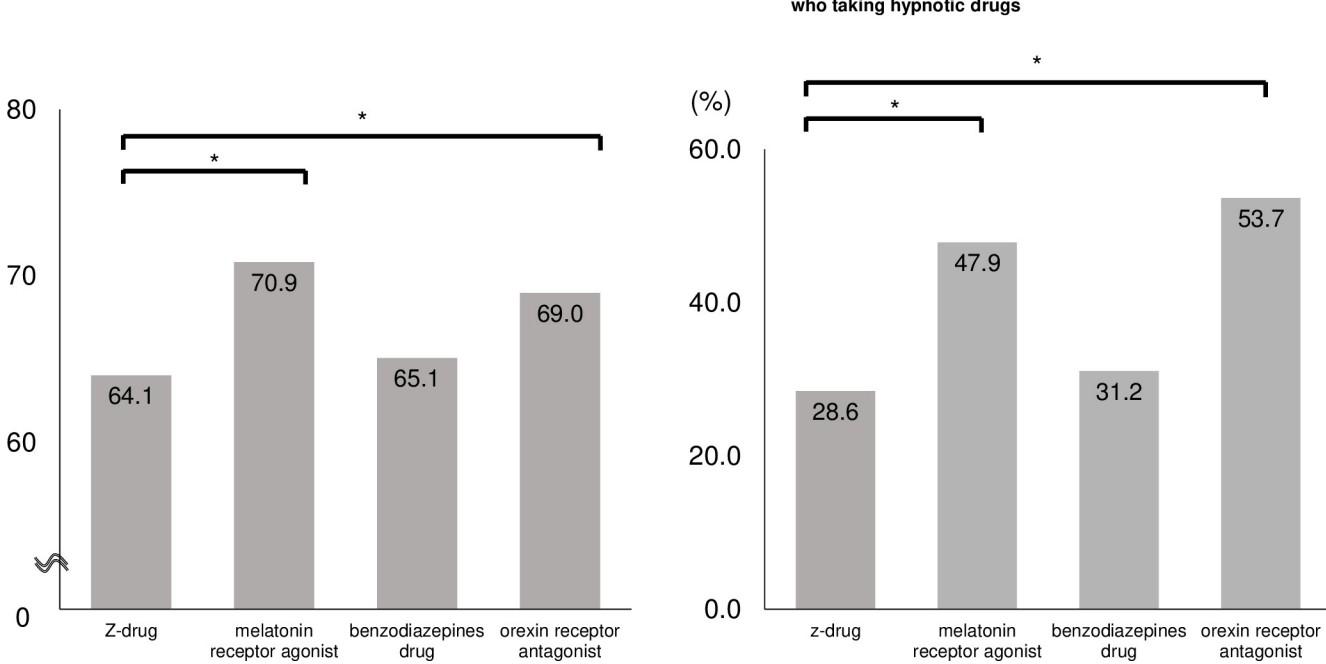

**Fig 4. Age and ratio of balance disorder and each patient who was prescribed hypnotic drugs.** The patients taking z-drugs were significantly younger than those taking melatonin receptor agonists and orexin receptor antagonists. The ratio balance disorder in the patients taking z-drugs was significantly smaller than that of melatonin receptor agonists and orexin receptor antagonists. *: means statistically significant.

odds ratio of falls in the univariate analysis (p = 0.0002). Benzodiazepines drugs and melatonin receptor agonists were not significantly associated with an increased odds ratio of falls in the univariate analysis (p = 0.4024, p = 0.9541) (Table 2).

The multivariate regression analysis was done with variables related to falls; age, sex, dosage, SIDE levels, the use of multiple drugs, angiotensin-converting enzyme inhibitors, alpha-blockers, beta-blockers, alpha beta-blockers, loop diuretics, and selective serotonin reuptake inhibitors. Z-drugs were significantly associated with a decreased odds ratio of falls in the multivariate analysis (p = 0.0135). Melatonin receptor agonists were not significantly associated with a decreased odds ratio of falls in the multivariate analysis (p = 0.24). Benzodiazepine drugs and melatonin receptor agonists were not significantly associated with an increased odds ratio of falls in the multivariate analysis (p = 0.089, p = 0.1128) (Table 2).

### Decision-tree algorithm for falls

Decision-tree algorithm was done with variables related to falls; age, gender, dosage, SIDE levels, the use of multiple drugs, and hypnotic drugs, angiotensin converting enzyme inhibitors, alpha blockers, beta blockers, alpha beta blockers, loop diuretics, and selective serotonin reuptake inhibitors. To clarify the profiles associated with falls, a decision tree algorithm was created using two divergence variables and classifying the patients into three groups (Fig 5). The SIDE levels were the initial divergence variable. Falls were seen in 2.9% of patients with normal balance ability (the SIDE levels 2b, 3, and 4). Among the patients with balance disorders (the SIDE level 0, 1, and 2a), gender was the variable for the second classification. Falls were seen in 11.4% of female patients. On the other hand, falls were seen in 17.8% of male patients. In this analysis, drugs weren't identified as divergence variables for falls.

## Discussion

Hypnotic drugs and balance are two causes of falls. In this study, we investigated the relationship between hypnotic drugs, balance and falls in hospitalized patients. Accidents involving falls were reported for 211 (7.4%) of 2860 inpatients during hospitalization. Z-drug recipients had the lowest fall rate among the hypnotic drugs. We analyzed to identify independent factors for falls. However, hypnotic drugs weren't independent factors of falls using the decision-tree algorithm. The SIDE balance assessment was the initial divergence variable. In order to prevent falls, it seems important not only to select appropriate hypnotic drugs but also to assess patients for balance and implement preventive measures.

**Table 2. Cox regression analysis for fall.**

| Factors | Univariate analysis for fall odds ratio (95% Confidence interval, P-value) | Multivariate analysis for fall odds ratio (95% Confidence interval, P-value) |
|---|---|---|
| Benzodiazepines drugs | 1.21 | 1.50 |
| | (0.76–1.89, 0.4024) | (0.94–2.39, 0.089) |
| Z-drugs | 0.57 | 0.69 |
| | (0.42–0.75, < .0001) | (0.52–0.93, 0.0135) |
| Melatonin receptor agonists | 1.01 | 0.78 |
| | (0.68–1.50, 0.9541) | (0.52–1.18, 0.24) |
| Orexin receptor antagonists | 1.76 | 1.28 |
| | (1.31–2.37, 0.0002) | (0.94–1.74, 0.1128) |

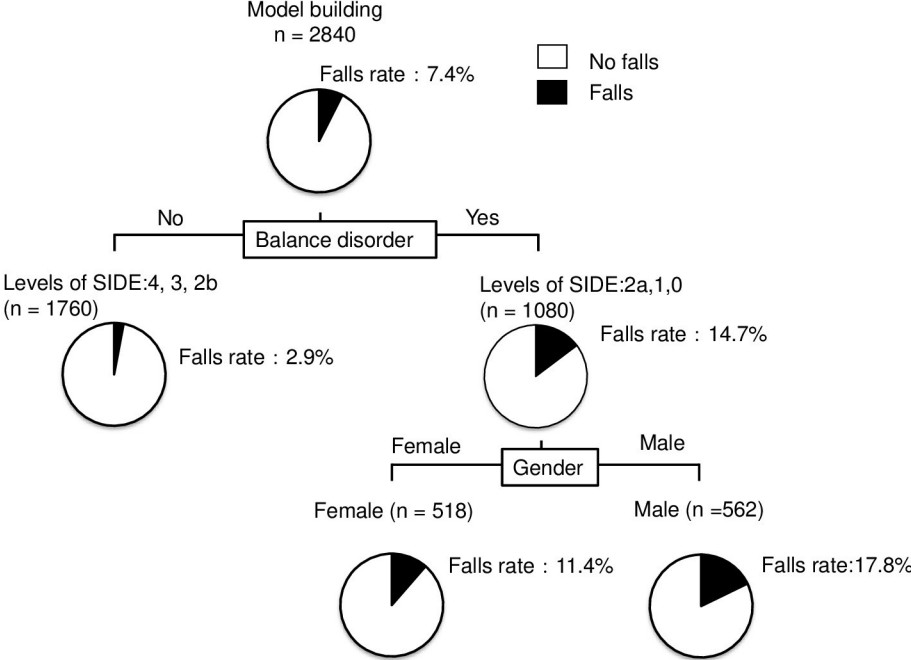

**Fig 5. Decision-tree algorithm for falls in hospitalized patients taking hypnotic drugs.** The pie indicates the proportion of patients with falls (black) and without falls (white). The falls rate in patients at SIDE level 2a or lower was five times the falls rate in patients at SIDE level 2b or higher. In addition, the falls rate in males was higher than in females for SIDE level 2a or lower.

## The relationship between falls rate and hypnotic drugs

Z-drugs were developed to overcome disadvantages of benzodiazepine drugs, including confusion, dependence, and abuse [17]. Z-drugs are safer and better as a first-line choice for treatment of insomnia than benzodiazepine drugs [18]. The falls rate for z-drugs was significantly lower than for benzodiazepine drugs in this study. Z-drugs were significantly associated with a decreased odds ratio of falls in the multivariate analysis. On the other hand, Ishibashi et al. have been reported that Z-drugs have a higher fall risk than benzodiazepines [19]. It remains unclear why these results are different. However, there are two possible reasons. First, our patients were younger than those of Ishibashi et al. In particular, the age of our patients who were taking oral z-drugs was 64.1 years which was younger than that of the patients of Ishibashi et al. Second, the study by Ishibashi et al. is a case-control study, with age- and gender-adjusted patients. They compared the results in age- and gender-adjusted patients. Their study included 44.3% and 67.9% of patients in the fall and control groups who were not taking hypnotic drugs. On the other hand, our study included only patients taking hypnotic drugs. These two differences may be related to the results regarding z-drugs. In this study, z-drugs resulted in the lowest fall rate, however, since all hypnotic drugs had some effect on falls in this study, the physician should consider the risk of falls when prescribing z-drugs as well.

Orexin receptor antagonists cause less postural instability than other hypnotic drugs, and do not impair alertness [20]. Orexin receptor antagonists are reported to be safe and effective hypnotic drugs for older patients in a systematic review [21]. Although the fall rate for orexin receptor antagonists was significantly higher than for z-drugs in this study, the causes remain unclear. Possible reasons include the following. Patients receiving orexin receptor antagonists were significantly older than those receiving z-drugs. Moreover, the ratio of balance disorder

was higher in patients with orexin receptor antagonists than with z-drugs. This trend has been reported in other studies. Ishibashi et al. [19] demonstrated that orexin receptor antagonists were associated with falls and were often prescribed to elderly patients. Our hospital recommends that physicians prescribe melatonin receptor agonists and orexin receptor antagonists, not benzodiazepines, to prevent falls. Therefore, clinicians may have prescribed orexin receptor antagonists to older patients with balance disorders when prescribing drugs for insomnia. Orexin receptor antagonists are safe drugs, but the risk of falls should be considered when prescribing them to elderly patients with a balance disorder.

## The independent causes of falls

In this study, we investigated the independent causes of falls. Balance disorders assessed using SIDE in decision tree analysis was an independent factor for causes of falls. The independent factor resulting in falls showed to be the SIDE level rather than the types of hypnotic drug in this analysis. This result may indicate that balance disorders need to be considered in addition to the choice of hypnotic drugs in order to prevent falls. The SIDE is a method devised to identify and assess the ability to maintain balance in a static standing position, allowing a simple assessment of patients immediately after admission to the ward [7]. Balance disorder was the initial divergence variable. The rate of falls in patients at SIDE level ≦ 2a was 14.7%. On the other hand, the rate of falls in patients at SIDE level≧2b was 2.9%. This result indicates the patients who can stand in a tandem position rarely fall. Furthermore, Teranishi et al. reported that no falls had been observed in patients at a SIDE level of 2b or higher in a rehabilitation hospital in a previous study [22]. This aligns with our findings. The SIDE may be a useful tool for assessing risk of falls in acute phase hospitals. Based on these results, while it is important to select hypnotics that have fewer side effects such as falls, it is also important to implement fall prevention in clinical practice by identifying high risk patients using balance assessment.

## Limitation

This study has several limitations. First, our study evaluated balance function in patients who took the hypnotic drug. We assessed patients' balance ability at admission in this study. On the other hand, the timing of hypnotic prescriptions varies. Some patients had begun taking hypnotics before being admitted, while others started them during hospitalization. This is a limitation of this retrospective study. A prospective randomized controlled trial is needed to resolve this issue. Second, we assessed only balance function and were unable to assess other risk factors such as the patient's muscle strength, walking, visible impairment, and previous falls.

## Conclusions

This study investigated the fall ratio of patients who took the hypnotic drug. The fall ratio for z-drugs was significantly lower than for benzodiazepine drugs, and orexin receptor antagonists. Moreover, we also investigated the factors associated with falls. Balance disorder was an independent factor for falls using decision-tree analysis. Therefore, a physician should consider not only the choice of drugs but also balance functions in inpatients.

## Supporting information

**S1 Table.**
(DOCX)

## Acknowledgments

We would like to thank Mr. Akihiro Ejima for cooperating with data acquisition.

## Author Contributions

**Data curation:** Ryuki Hashida, Hiroshi Tajma, Masafumi Bekki, Sohei Iwanaga, Koji Hara, Yosuke Nakamura, Yuji Kaneyuki.

**Formal analysis:** Ryuki Hashida, Shinji Yokoyama, Sohei Iwanaga.

**Investigation:** Ryuki Hashida, Shinji Yokoyama, Sayuri Kawano, Eriko Higashi, Hiroshi Tajma, Masafumi Bekki, Sohei Iwanaga, Koji Hara, Yosuke Nakamura, Yuji Kaneyuki.

**Project administration:** Sayuri Kawano.

**Supervision:** Hiroo Matsuse, Takeshi Nago, Yoshihiro Fukumoto, Motohiro Ozone, Naohisa Uchimura, Naoto Shiba.

**Writing – original draft:** Ryuki Hashida.

**Writing – review & editing:** Hiroo Matsuse, Shinji Yokoyama, Eriko Higashi, Hiroshi Tajma, Masafumi Bekki, Sohei Iwanaga, Koji Hara, Yosuke Nakamura, Yuji Kaneyuki, Takeshi Nago, Yoshihiro Fukumoto, Motohiro Ozone, Naohisa Uchimura, Naoto Shiba.

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
