## [Decision Letter · Decision Letter 0]

18 Apr 2022

PONE-D-22-02308Balance dysfunction the most significant cause of in-hospital falls in patients taking hypnotic drugs: A retrospective studyPLOS ONE

Dear Dr. Hashida,

Thank you for submitting your manuscript to PLOS ONE. After careful consideration, we feel that it has merit but does not fully meet PLOS ONE’s publication criteria as it currently stands. Therefore, we invite you to submit a revised version of the manuscript that addresses the points raised during the review process.

Overall, the manuscript needs major revisions before being considered for publication. Specially:

The manuscript needs to be more detailed to clarify the study design, study gap and  clinical importance of using decision trees .I would also recommend proofreading the manuscript before submission for spellings and grammar.

We look forward to receiving your revised manuscript.

Kind regards,

Amir Radfar, MD,MPH,MSc,DHSc

Academic Editor

PLOS ONE

Journal Requirements:

[Unfunded studies]. 

Reviewers' comments:

Reviewer's Responses to Questions

**Comments to the Author**

1. Is the manuscript technically sound, and do the data support the conclusions?

Reviewer #1: Partly

Reviewer #2: Yes

2. Has the statistical analysis been performed appropriately and rigorously? 

Reviewer #1: No

Reviewer #2: Yes

3. Have the authors made all data underlying the findings in their manuscript fully available?

Reviewer #1: Yes

Reviewer #2: Yes

4. Is the manuscript presented in an intelligible fashion and written in standard English?

Reviewer #1: Yes

Reviewer #2: No

5. Review Comments to the Author

Reviewer #1: Although I believe that this study is based on good research question, several critical problems and questions exist.

1. it is not clear why the analysis was conducted using decision trees; if the goal is to evaluate SIDE and sleeping pills, why not just include SIDE and sleeping pills at the same time in the multivariate regression analysis? (in case-control study or cross sectional study)

It may be necessary to clarify the clinical importance of using decision trees in the background and discussion.

2. Are there any reasons for limiting the study participants who have been prescribed hypnotic drugs?

When examining the effects of SIDE and hypnotic drugs, limiting the study to those who took hypnotic drugs not only reduces the external validity of this study, but also causes selection bias.

Reasonable rationale for the selection of study subjects is necessary.

3. Isn't it necessary to consider age in the presentation of results?

I consider that it is necessary to take age into account when comparing the ability of balance between patients prescribing hypnotic drugs.

In general, melatonin receptor agonists tend to be used in elderly patients in Japan.

This may lead to more balance problems in patients who prescribing melatonin receptor agonists.

Is the decision tree adjusted for age?

If so, it is necessary to specify in what form the variable is included (continuous or category like more than 65 years old or not).

4. Z-drug users had the lowest fall rate among the hypnotics, but is there any effect of age or other factors?

In general, Z-drugs have been reported to have a higher fall risk than benzodiazepines (Ishibashi, Y. et al. PloS one, 15(9), e0238723. Brandt, J., & Leong, C. (2017). Drugs in R & D, 17(4), 493-507.)

Differences from previous studies should be described in the discussion.

5. It needs to be clearly stated how the study dealt with patients who took multiple hypnotic drugs.

The authors stated, "We investigated the top 6 hypnotic drugs that were often prescribed as follows,". Only the most commonly prescribed drugs in patients were used as variables, and the others were ignored? This needs to be described in methods.

It should be stated whether the dosage was also taken into account.

6. It is necessary to clarify whether the high risk of falling at orexin receptors is due to age or to low balance ability.

According to previous study (Ishibashi, Y. et al. PloS one, 15(9), e0238723), orexin receptors are reported to have a higher risk of falls.

However, they are more likely to be administered to the elderly in Japan, and people with frail may be administered them.

7. More details are needed on the timing of SIDE assessment.

Was the SIDE taken prior to admission?

Was the patient taking medication internally at that time?

It needs to be clarified whether the SIDE represents a loss of balance due to medication or whether it represents the patient's original degree of frailty.

Was the SIDE assessed on admission, followed by the start of drug medication and the assessment of falls?

Related to the above, the timing of the collection of the patient's exposure of oral medications also needs to be described.

Are the medications taken into account when the fall occurred? Or are you considering oral medication upon admission? Do you take into account the medication at discharge?

Do you need to consider the possibility that the medication may change due to falls?

Differences in the timing of medication prescribing between patients who have fallen and those who have not fallen may be a source of bias.

Reviewer #2: Thank you for your interesting paper.

- Abstract:

o Concise and readable.

- Introduction:

o Please describe that why patients with a hospital-acquired hip fracture have higher complication rate than those who fall outside of the hospital.

o The study gap is not clearly stated in the introduction section. The authors can highlight the components of the standing balance postural controls in elderly subjects.

- Methods

o There is a discrepancy in duration of data gathering. In line 122, the authors stated that data gathering was conducted from July to December 2021. However, in line 128 (for figure legend), it was reported as July to December 2020. Please check that which one is correct.

o In introduction section, the authors stated that in their hospital benzodiazepine in not prescribed for patients. However, in method section the benzodiazepine is one of the hypnotic drugs that prescribed to the patients.

o Line 204: please report the p-value for fall ratio between z-drugs and melatonin receptor agonists.

- Discussion:

o Please compare the results of this study with related articles that published previously.

o In limitation section, please describe that why retrospective nature of the study design is one of limitation of the study.

- Conclusion:

o Relevant to the main findings.

- General point:

o There is some grammatical error in the text that should be revised by a native fluent reviewer.

6. PLOS authors have the option to publish the peer review history of their article (what does this mean?). If published, this will include your full peer review and any attached files.

Reviewer #1: **Yes: **Yoshiki Ishibashi

Reviewer #2: **Yes: **Taher Babaee

---

## [Author Response · Author response to Decision Letter 0]

3 Jun 2022

Responses to REVIEWER 1:

Thank you for your comments regarding our manuscript (PONE-D-22-02308). We appreciate your comments, which have helped us to improve our manuscript. In line with your comments, please find below our point-by-point responses.

Specific comments

Introduction

Comment 1:

it is not clear why the analysis was conducted using decision trees; if the goal is to evaluate SIDE and sleeping pills, why not just include SIDE and sleeping pills at the same time in the multivariate regression analysis? (in case-control study or cross-sectional study) It may be necessary to clarify the clinical importance of using decision trees in the background and discussion.

Answer: We appreciate your valuable advice to improve our research. We added some sentences about decision tree analysis in the statistical analysis section. We used decision tree analysis to identify the most influential factors associated with falls and the combinations of factors for falls. We also conducted a cox regression analysis. The multivariate regression analysis was done with variables related to falls; age, sex, dosage, Standing Test for Imbalance and Disequilibrium levels, the use of multiple drugs, angiotensin-converting enzyme inhibitors, alpha-blockers, beta-blockers, alpha beta-blockers, loop diuretics, and selective serotonin reuptake inhibitors were significant factors for falling. We added these results in the result section and the discussion section. 

(Page7, line183-184) (Page10, line241-257) 

Table2. Cox regression analysis for fall

Factors Univariate analysis for fall OR

 (95% Confidence interval, P-value) Multivariate analysis for fall OR

 (95% Confidence interval, P-value)

Benzodiazepines drugs 1.21 

(0.76-1.89, 0.4024) 1.50 

(0.94-2.39, 0.089)

Z-drugs 0.57 

(0.42-0.75, <.0001) 0.69 

(0.52-0.93, 0.0135)

Melatonin receptor agonists 1.01 

(0.68–1.50, 0.9541) 0.78 

(0.52-1.18, 0.24)

Orexin receptor antagonists 1.76 

(1.31–2.37, 0.0002) 1.28 

(0.94-1.74, 0.1128)

Comment 2: Are there any reasons for limiting the study participants who have been prescribed hypnotic drugs? When examining the effects of SIDE and hypnotic drugs, limiting the study to those who took hypnotic drugs not only reduces the external validity of this study, but also causes selection bias. Reasonable rationale for the selection of study subjects is necessary.

Answer: We appreciate your valuable comment. Hypnotic drugs impair balance ability [1]. Therefore, we need to pay attention to the balance function of patients taking hypnotic drugs. There was no study focused on investigating falls with patients taking hypnotic drugs. For this reason, the title stated that the patient was taking hypnotic drugs internally. However, as you pointed out, the introduction was inadequately written and did not provide enough background regarding the fact that the study subjects were patients taking hypnotic drugs. We revised the introduction to clarify this study's aim. 

(Page4, line76 – Page5, line114)

Comment 3: Isn't it necessary to consider age in the presentation of results?

I consider that it is necessary to take age into account when comparing the ability of balance between patients prescribing hypnotic drugs. In general, melatonin receptor agonists tend to be used in elderly patients in Japan.　This may lead to more balance problems in patients who prescribing melatonin receptor agonists. Is the decision tree adjusted for age? If so, it is necessary to specify in what form the variable is included (continuous or category like more than 65 years old or not).

Answer: 

We apologize for our unclear description. The age of patients taking melatonin receptor agonists was higher than that of other patients (Figure 4A). Moreover, the ratio of balance disorder in patients taking melatonin receptor agonists was higher than that of patients taking other drugs (Figure4B). As you noted, aging impairs balance functions, so we thought we should adjust for the effects of age as well. Decision-tree algorithm was done with variables related to falls; age, sex, SIDE levels, the use of multiple drugs, hypnotic drugs, angiotensin-converting enzyme inhibitors, alpha-blockers, beta-blockers, alpha beta-blockers, loop diuretics, and selective serotonin reuptake inhibitors. Age was not a significant factor in falls in the decision tree analysis. (Page11, line 263-266) 

Moreover, we have added the results of the Cox regression analysis. The melatonin receptor agonists were not significantly associated with an increased odds ratio of falls in cox regression analysis. We added these results in the main texts. Thank you for your advice.

(Page10, line 253-255) 

Comment 4: Z-drug users had the lowest fall rate among the hypnotics, but is there any effect of age or other factors? In general, Z-drugs have been reported to have a higher fall risk than benzodiazepines (Ishibashi, Y. et al. PloS one, 15(9), e0238723. Brandt, J., & Leong, C. (2017). Drugs in R & D, 17(4), 493-507.) Differences from previous studies should be described in the discussion.

Answer: 

Thank you for your valuable comments. As you pointed out, there are some reports using z-drug-caused falls [2, 3]. However, the muscle-relaxant effect of z-drugs is weaker than benzodiazepines; the risk of falls with z-drugs is low [4]. Thus, we consider that no certain conclusion has been reached regarding the relationship between z-drugs and falls. Comparing the present study and results with those reported by Ishibashi et al. [5], the two points are different. First, our patients were younger than those of Ishibashi et al. In particular, the age of our patients who were taking oral z-drugs was 64.1 years which was younger than that of the patients of Ishibashi et al. In this study, attending physicians may have prescribed an orexin receptor agonist to elderly patients with impaired balance function (Figure 4 A, B). Second, the study by Ishibashi et al. is a case-control study, with age- and gender-adjusted patients. They compared the results in age- and gender-adjusted patients. Their study included 44.3% and 67.9% of patients in the fall and control groups who were not taking hypnotic medication. On the other hand, our study included only patients taking hypnotic medication. These two differences may be related to the results regarding z-drugs. We added these sentences in the discussion sections. 

(Page12, line296 - Page13, line315)

Comment 5: It is necessary to clarify whether the high risk of falling at orexin receptors is due to age or to low balance ability. According to previous study (Ishibashi, Y. et al. PloS one, 15(9), e0238723), orexin receptors are reported to have a higher risk of falls. However, they are more likely to be administered to the elderly in Japan, and people with frail may be administered them.

Answer: We fully agree with you. The patients who took orexin receptor antagonists were elderly and had compromised balance (Figure 4 A, B). We performed an additional multivariate analysis on patients taking oral orexin receptor antagonists. These patients' falls were associated with a balance disorder. Please see the supplemental table below. In a systematic review, orexin receptor antagonists are reported to be safe and effective hypnotic drugs for older patients [6]. However, Ishibashi et al. [5] demonstrated that orexin receptor antagonists were associated with falls and were often prescribed to elderly patients. Taken together, orexin receptor antagonists are safe drugs, but the risk of falls should be considered when prescribing them to elderly patients with a balance disorder. We added this point in the discussion section. Thank you for your advice.

(Page13, line 325-327)

Supplementally table1: Cox regression analysis for falls in patients who took orexin receptor antagonists

Factors OR

 (95% Confidence interval, P-value)

age 1.01 

(0.10–1.03, 0.1198)

Balance disorder 5.41 

(2.87–10.19, <.0001)

Multiple logistic regression adjusted age, standing test for Imbalance and disequilibrium levels

Comment 6: It needs to be clearly stated how the study dealt with patients who took multiple hypnotic drugs. The authors stated, "We investigated the top 6 hypnotic drugs that were often prescribed as follows,". Only the most commonly prescribed drugs in patients were used as variables, and the others were ignored? This needs to be described in methods. It should be stated whether the dosage was also taken into account.

Answer: We apologize for the inadequate description of our methodology. We extracted patient data from inpatient prescription records of hypnotic drugs. Therefore, the number of patients is the total number of patients. We also investigated the dosage of each patient. We also included dosage as a variable in the multivariate analysis. We have added sentences in the methods section.

(Page6, line141-143)

Comment 7: More details are needed on the timing of SIDE assessment. Was the SIDE taken prior to admission?　Was the patient taking medication internally at that time?　It needs to be clarified whether the SIDE represents a loss of balance due to medication or whether it represents the patient's original degree of frailty. Was the SIDE assessed on admission, followed by the start of drug medication and the assessment of falls?　Related to the above, the timing of the collection of the patient's exposure of oral medications also needs to be described.　Are the medications taken into account when the fall occurred? Or are you considering oral medication upon admission? Do you take into account the medication at discharge?　Do you need to consider the possibility that the medication may change due to falls?　Differences in the timing of medication prescribing between patients who have fallen and those who have not fallen may be a source of bias.

Answer: We assessed the patient's balance ability when they were admitted. On the other hand, the timing of the prescription of the hypnotic drug is different. Some patients who are admitted to the hospital have already been taking some hypnotic drugs, while others start during their hospital stay. Regarding the timing of falling and medication, as you pointed out, we were not able to take into account the timing when the fall occurred. As you pointed out, this point is a limitation of this retrospective study. A prospective randomized control study is needed to solve this problem. We added this point in the limitation section. Thank you for your advice.

(Page14, line356-364)

1. Bland H, Li X, Mangin E, Yee KL, Lines C, Herring WJ, et al. Effects of Bedtime Dosing With Suvorexant and Zolpidem on Balance and Psychomotor Performance in Healthy Elderly Participants During the Night and in the Morning. J Clin Psychopharmacol. 2021;41(4):414-20. Epub 2021/06/29. doi: 10.1097/JCP.0000000000001439. PubMed PMID: 34181362.

2. Bellazzi R, Zupan B. Predictive data mining in clinical medicine: current issues and guidelines. Int J Med Inform. 2008;77(2):81-97. Epub 2006/12/26. doi: 10.1016/j.ijmedinf.2006.11.006. PubMed PMID: 17188928.

3. Kolla BP, Lovely JK, Mansukhani MP, Morgenthaler TI. Zolpidem is independently associated with increased risk of inpatient falls. J Hosp Med. 2013;8(1):1-6. Epub 2012/11/21. doi: 10.1002/jhm.1985. PubMed PMID: 23165956.

4. Obayashi K, Araki T, Nakamura K, Kurabayashi M, Nojima Y, Hara K, et al. Risk of falling and hypnotic drugs: retrospective study of inpatients. Drugs R D. 2013;13(2):159-64. Epub 2013/06/14. doi: 10.1007/s40268-013-0019-3. PubMed PMID: 23760758; PubMed Central PMCID: PMCPMC3689908.

5. Ishibashi Y, Nishitani R, Shimura A, Takeuchi A, Touko M, Kato T, et al. Non-GABA sleep medications, suvorexant as risk factors for falls: Case-control and case-crossover study. PLoS One. 2020;15(9):e0238723. Epub 2020/09/12. doi: 10.1371/journal.pone.0238723. PubMed PMID: 32916693; PubMed Central PMCID: PMCPMC7486134 Chiba, Keiko Ashidate, Tomoyasu Ichijo, Takanori Shirai, Nobuo Ishiwata, Masataka Sasabe. declares that they have no conflicts of interest. Yoshiki Ishibashi, reports personal fees from Children and Future co., Ltd. Akiyoshi Shimura reports personal fees from Dainippon Sumitomo Pharma, Yoshitomiyakuhin, and Meiji Seika Pharma. This does not alter our adherence to PLOS ONE policies on sharing data and materials.

6. Sys J, Van Cleynenbreugel S, Deschodt M, Van der Linden L, Tournoy J. Efficacy and safety of non-benzodiazepine and non-Z-drug hypnotic medication for insomnia in older people: a systematic literature review. Eur J Clin Pharmacol. 2020;76(3):363-81. Epub 2019/12/16. doi: 10.1007/s00228-019-02812-z. PubMed PMID: 31838549.

 

Responses to REVIEWER 2:

Thank you for your comments regarding our manuscript (PONE-D-22-02308). We appreciate your comments, which have helped us to improve our manuscript. In line with your comments, please find below our point-by-point responses.

Specific comments

Introduction

Comment 1:

Please describe that why patients with a hospital-acquired hip fracture have higher complication rate than those who fall outside of the hospital.

Answer: We fully agree with your comment. We added the reason why the patients with a hospital-acquired hip fracture have a poor prognosis. The patients with a hospital-acquired hip fracture are already in poor condition physically and mentally and have an especially poor outcome [1]. We added this sentence in the introduction sections. Thank you for your advice.

(Page4, line83-85)

Comment 2: The study gap is not clearly stated in the introduction section. The authors can highlight the components of the standing balance postural controls in elderly subjects.

Answer: We apologize for our unclear description. We should have clarified the study gap in the introduction section. Hypnotic drugs, including z-drug and orexin receptor antagonists, affect balance ability [2]. Therefore, we need to pay attention to the balance function of patients taking hypnotic drugs. We revised the introduction section to clarify this study’s aim in the introduction sections.

(Page5, line111-114)

Comment 3: o There is a discrepancy in the duration of data gathering. In line 122, the authors stated that data gathering was conducted from July to December 2021. However, in line 128 (for figure legend), it was reported as July to December 2020. Please check that which one is correct.

Answer: We apologize for the typo. I wrote the following sentence. Thank you for your careful reading. We did a retrospective study and enrolled all hospitalized patients who were taking hypnotic drugs from July to December 2020 at Kurume University Hospital.

(Page5, line126-128)

Comment 4: In introduction section, the authors stated that in their hospital benzodiazepine in not prescribed for patients. However, in method section the benzodiazepine is one of the hypnotic drugs that prescribed to the patients.

Answer: Thank you for pointing out the unclear points in our sentences. Our hospital recommends that physicians not prescribe benzodiazepines in case of insomnia. However, there were patients who had been taking benzodiazepines before they were admitted to our hospital in this study. As you point out, the introduction section seems misleading. We removed this sentence and revised the introduction sentences. 

Comment 5: Line 204: please report the p-value for fall ratio between z-drugs and melatonin receptor agonists.

Answer: Thank you for your careful reading. As you suggested, we wrote the p-value for fall ratio between z-drugs and melatonin receptor agonists. We added the following sentence in the results sections. 

However, there was no significant difference in the fall ratio between z-drugs and melatonin receptor agonists (p=0.1352).

(Page9, line213-214)

Comment 6: Please compare the results of this study with related articles that published previously.

Answer: Thank you for your advice. We compared the results of each drug fall rate in this study with the previous study in the discussion section. I appreciate your valuable comments.

(Page12, line300- Page13, line315)

Comment 7: In limitation section, please describe that why retrospective nature of the study design is one of limitation of the study.

Answer: Thank you for pointing this out. Our study evaluated balance function in patients who took the hypnotic drug. We assessed patients' balance ability at admission in this study. On the other hand, the timing of hypnotic prescriptions varies. Some patients had begun taking hypnotics before being admitted, while others started them during hospitalization. This is a limitation of this retrospective study. A prospective randomized controlled trial is needed to resolve this issue. We have added this point to the limitations section.

 (Page14, line356-364)

Comment 8: - Conclusion: Relevant to the main findings.

Answer: We appreciate your comments. As you pointed out, the conclusion was inadequate. We modified the conclusion according to the results as follows. 

This study investigated the ratio of patients who took the hypnotic drug. The falls ratio for z-drugs was significantly lower than for benzodiazepine drugs, and orexin receptor antagonists. Moreover, we also investigated the factors associated with falls. Balance disorder was an independent factor for falls using decision-tree analysis. Therefore, a physician should consider not only the choice of drugs but also balancing functions in inpatients.

(Page14, line368-373)

Comment 9: General point: There is some grammatical error in the text that should be revised by a native fluent reviewer.

Answer: Thank you for your advice. A native speaker had checked this manuscript before submission. We asked her to recheck this manuscript after revision. We consider that this manuscript's grammar has improved. 

1. Foss NB, Palm H, Kehlet H. In-hospital hip fractures: prevalence, risk factors and outcome. Age Ageing. 2005;34(6):642-5. Epub 2005/11/04. doi: 10.1093/ageing/afi198. PubMed PMID: 16267194.

2. Bland H, Li X, Mangin E, Yee KL, Lines C, Herring WJ, et al. Effects of Bedtime Dosing With Suvorexant and Zolpidem on Balance and Psychomotor Performance in Healthy Elderly Participants During the Night and in the Morning. J Clin Psychopharmacol. 2021;41(4):414-20. Epub 2021/06/29. doi: 10.1097/JCP.0000000000001439. PubMed PMID: 34181362.

---

## [Decision Letter · Decision Letter 1]

7 Jul 2022

PONE-D-22-02308R1Balance dysfunction the most significant cause of in-hospital falls in patients taking hypnotic drugs: A retrospective studyPLOS ONE

Dear Dr. Hashida,

Thank you for submitting your manuscript to PLOS ONE. After careful consideration, we feel that it has merit but does not fully meet PLOS ONE’s publication criteria as it currently stands. Therefore, we invite you to submit a revised version of the manuscript that addresses the points raised during the review process.

Specifically:    More than 50 percent of the references used in this manuscript are over 5 years. Except for “seminal” works, please use updated references.   Please address comments made by Reviewer #3

We look forward to receiving your revised manuscript.

Kind regards,

Amir Radfar, MD,MPH,MSc,DHSc

Academic Editor

PLOS ONE

Journal Requirements:

Reviewers' comments:

Reviewer's Responses to Questions

**Comments to the Author**

1. If the authors have adequately addressed your comments raised in a previous round of review and you feel that this manuscript is now acceptable for publication, you may indicate that here to bypass the “Comments to the Author” section, enter your conflict of interest statement in the “Confidential to Editor” section, and submit your "Accept" recommendation.

Reviewer #2: All comments have been addressed

Reviewer #3: All comments have been addressed

2. Is the manuscript technically sound, and do the data support the conclusions?

Reviewer #2: Yes

Reviewer #3: Yes

3. Has the statistical analysis been performed appropriately and rigorously? 

Reviewer #2: Yes

Reviewer #3: Yes

4. Have the authors made all data underlying the findings in their manuscript fully available?

Reviewer #2: Yes

Reviewer #3: Yes

5. Is the manuscript presented in an intelligible fashion and written in standard English?

Reviewer #2: Yes

Reviewer #3: Yes

6. Review Comments to the Author

Reviewer #2: Dear authors,

Thank you very much for your thorough revision of the present form of above-mentioned manuscript.

Reviewer #3: All the reviewers comments are addressed. However, I would like to ask about figure 6 is balance disorder " no" when level of SIDE is :2a,1,0?

Also, the aim of work in abstract: The aim of this study was to investigate the cause of falls in hospitalized patients taking hypnotic drugs, while, in the introduction section, The purpose of this study was to investigate the fall rate of each patient who took the hypnotic drug and the factor associated with falls.

in the conclusion section, "This study investigated the ratio of patients who took the hypnotic drug" is this accurate as regards the results?

7. PLOS authors have the option to publish the peer review history of their article (what does this mean?). If published, this will include your full peer review and any attached files.

Reviewer #2: **Yes: **Taher Babaee

Reviewer #3: No

---

## [Author Response · Author response to Decision Letter 1]

9 Jul 2022

Responses to REVIEWER 3:

Thank you for your comments regarding our manuscript (PONE-D-22-02308R1). We appreciate your comments, which have helped us to improve our manuscript. In line with your comments, please find below our point-by-point responses.

Specific comments

Comment 1:

All the reviewers’ comments are addressed. However, I would like to ask about figure 6 is balance disorder " no" when the level of SIDE is:2a,1,0?

Answer: I apologize for my miswriting in figure 6. We defined patients who were unable to do tandem standing as having balance disorder (SIDE levels 0, 1, and 2a) (Page6, line153-155). We revised figure 6, thank you for your careful reading.

Comment 2: Also, the aim of work in abstract: The aim of this study was to investigate the cause of falls in hospitalized patients taking hypnotic drugs, while, in the introduction section, The purpose of this study was to investigate the fall rate of each patient who took the hypnotic drug and the factor associated with falls. In the conclusion section, "This study investigated the ratio of patients who took the hypnotic drug" is this accurate as regards the results?.

Answer: As you pointed out, the aim in the abstract and introduction was not appropriate. I revised the aim in the abstract section as follows. “The purpose of this study was to investigate the fall rate of each patient who took the hypnotic drug and the factor associated with falls.” (Page3, line51-53) Moreover, I miswrote the conclusion in the main text. I revised the conclusion as follows.” This study investigated the fall ratio of patients who took the hypnotic drug. The fall ratio for z-drugs was significantly lower than for benzodiazepine drugs, and orexin receptor antagonists. Moreover, we also investigated the factors associated with falls. Balance disorder was an independent factor for falls using decision-tree analysis. Therefore, a physician should consider not only the choice of drugs but also balance functions in inpatients.” (Page14, line369-374) I appreciate your valuable comments.

---

## [Decision Letter · Decision Letter 2]

28 Jul 2022

Balance dysfunction the most significant cause of in-hospital falls in patients taking hypnotic drugs: A retrospective study

PONE-D-22-02308R2

Dear Dr. Hashida,

We’re pleased to inform you that your manuscript has been judged scientifically suitable for publication and will be formally accepted for publication once it meets all outstanding technical requirements.

Kind regards,

Amir Radfar, MD,MPH,MSc,DHSc

Academic Editor

PLOS ONE

Additional Editor Comments (optional):

Reviewers' comments:

Reviewer's Responses to Questions

**Comments to the Author**

1. If the authors have adequately addressed your comments raised in a previous round of review and you feel that this manuscript is now acceptable for publication, you may indicate that here to bypass the “Comments to the Author” section, enter your conflict of interest statement in the “Confidential to Editor” section, and submit your "Accept" recommendation.

Reviewer #3: All comments have been addressed

Reviewer #4: All comments have been addressed

2. Is the manuscript technically sound, and do the data support the conclusions?

Reviewer #3: Yes

Reviewer #4: Yes

3. Has the statistical analysis been performed appropriately and rigorously? 

Reviewer #3: Yes

Reviewer #4: Yes

4. Have the authors made all data underlying the findings in their manuscript fully available?

Reviewer #3: Yes

Reviewer #4: Yes

5. Is the manuscript presented in an intelligible fashion and written in standard English?

Reviewer #3: Yes

Reviewer #4: Yes

6. Review Comments to the Author

Reviewer #3: Thanks for addressing the comments and for revising the manuscript again.

Thank you for this good research.

Reviewer #4: Thank you for the revised manuscript. I believe the authors addressed all questions during the review process . I recommend the manuscript for publication.

7. PLOS authors have the option to publish the peer review history of their article (what does this mean?). If published, this will include your full peer review and any attached files.

Reviewer #3: No

Reviewer #4: **Yes: **Irina Filip MD, DHSc

---

## [Editor Report · Acceptance letter]

23 Aug 2022

PONE-D-22-02308R2 

Balance dysfunction the most significant cause of in-hospital falls in patients taking hypnotic drugs: A retrospective study 

Dear Dr. Hashida:

I'm pleased to inform you that your manuscript has been deemed suitable for publication in PLOS ONE. Congratulations! Your manuscript is now with our production department. 

Kind regards, 

on behalf of

Dr. Amir Radfar 

Academic Editor

PLOS ONE